# Hundreds-Dollar-Level Multiplex Integrated RT-qPCR Quantitative System for Field Detection

**DOI:** 10.3390/bios12090706

**Published:** 2022-09-01

**Authors:** Zhihao Lan, Yu Guo, Kangning Wang, Yipeng Zhang, Youyun Chen, Dezhou Zheng, Xiaolong Xu, Wenming Wu

**Affiliations:** 1Institute of Biological and Medical Engineering, Guangdong Academy of Sciences, Guangzhou 510075, China; 2School of Biomedical Engineering, Southern Medical University, Guangzhou 510515, China; 3School of Mechanical and Electrical Engineering, Guangdong University of Technology, Guangzhou 510006, China; 4College of Applied Physics and Materials, Wuyi University, Jiangmen 529000, China; 5School of Biotechnology and Health Sciences, Wuyi University, Jiangmen 529000, China; 6State Key Laboratory of ASIC and Systems, Fudan University, Shanghai 200433, China

**Keywords:** real-time PCR, on-site testing, low cost, COVID-19, ASF, fluorescence detection

## Abstract

The COVID-19 pandemic poses a threat to global health. Due to its high sensitivity, specificity, and stability, real-time fluorescence quantitative (real-time PCR) detection has become the most extensively used approach for diagnosing SARS-CoV-2 pneumonia. According to a report from the World Health Organization, emerging and underdeveloped nations lack nucleic acid detection kits and polymerase chain reaction (PCR) instruments for molecular biological detection. In addition, sending samples to a laboratory for testing may result in considerable delays between sampling and diagnosis, which is not favorable to the timely prevention and control of new crown outbreaks. Concurrently, there is an urgent demand for accurate PCR devices that do not require a laboratory setting, are more portable, and are capable of completing testing on-site. Hence, we report on HDLRT-qPCR, a new, low-cost, multiplexed real-time fluorescence detection apparatus that we have developed for on-site testing investigations of diverse diseases in developing nations. This apparatus can complete on-site testing rapidly and sensitively. The entire cost of this instrument does not exceed USD 760. In order to demonstrate the applicability of our PCR instrument, we conducted testing that revealed that we achieved gradient amplification and melting curves comparable to those of commercially available equipment. Good consistency characterized the testing outcomes. The successful detection of target genes demonstrates the reliability of our inexpensive PCR diagnostic technique. With this apparatus, there is no need to transport samples to a central laboratory; instead, we conduct testing at the sampling site. This saves time on transportation, substantially accelerates overall testing speed, and provides results within 40 min.

## 1. Introduction

Polymerase chain reaction (PCR) technology is a method of amplifying target DNA sequences in vitro. DNA sequences can be amplified by several orders of magnitude through the use of this method, leading to thousands or even millions of copies of a single DNA sequence [1]. PCR technology is considered to be one of the greatest and most widely used inventions in medical and biological research [2,3,4].

The global COVID-19 pandemic resulting from the severe acute respiratory syndrome coronavirus 2 (SARS-CoV-2) has currently infected over 518 million people, and has taken the lives of over 6 million, resulting in a tremendous strain on health care services worldwide [5]. At the same time, there exist routine infectious diseases such as hepatitis B and tuberculosis that pose great hidden dangers to human life and health [6,7,8,9,10]. Currently, underdeveloped regions and countries with fragile medical and health systems have become the hardest-hit areas as a result of the COVID-19 epidemic. These regions typically have a high population density, a large proportion of poor people, and insufficient medical facilities and capabilities, all of which make epidemic prevention difficult [11,12].

With the sudden outbreak of infectious diseases around the world, making timely diagnoses is particularly important. Large-scale diagnostic testing is a key tool in epidemiology and in containing outbreaks such as COVID-19. These techniques have been the focus of international research [13,14,15,16]. Experts agree that reverse transcription-polymerase chain reaction (PCR) testing is crucial to the control of SARS-CoV-2 [15]. More rapid antigen tests as well as real-time polymerase chain reactions for on-site diagnoses have played a significant role in containing the spread of various infectious diseases. Increasing the efficiency of nucleic acid screening is a key objective. Sending samples to a central laboratory has some disadvantages, particularly with the long delays involved between sampling and diagnosis; in some cases, immediate testing at the point of sampling is urgently required. For example, PCR amplification testing is performed on-site at the sampling point, and the test results are also checked on-site. If SARS-CoV-2 is discovered, the patient can be located immediately, isolated and treated to prevent the disease from spreading.

However, the cost of these commercial quantitative PCR devices is relatively high, which is an impediment for resource-limited, underdeveloped regions. Furthermore, medical equipment is relatively outdated and there are few available doctors in these countries, thus they cannot cope well with sudden outbreaks as a result of inadequate resources [17]. The provision of on-site diagnostics in a pharmacy or community center is especially important at this time, and could save many lives. The demand for low-cost, real-time diagnostic equipment is becoming increasingly apparent. Remote and poor areas can greatly benefit from the use of on-site diagnostic devices, as they can greatly increase the efficiency of health care, as well as reduce the financial burden of health care in underdeveloped regions. Meanwhile, animal husbandry is a pillar industry in numerous developing nations, and is their economic backbone. Thus, we investigated the viability of our developed quantitative real-time PCR system application in this field. In animal husbandry, the polymerase chain reaction (PCR) has been widely implemented; moreover, the qualification approval and application process is faster in this field, and our instruments can be quickly put into use in underdeveloped areas to facilitate economic development of the livestock industry. African swine fever (ASF) is a highly contagious viral disease in swine that causes a high mortality rate, approaching 100%, in domestic pigs. It results in important economic losses that are unavoidable in the absence of an effective vaccine. The current practices for disease control include quarantining of the affected area as well as the slaughter of infected animals [18].

At present, PCR instruments mainly include conventional PCR, gradient PCR, in situ PCR, and quantitative real-time PCR (qPCR). Regular PCR and qPCR instruments are the most widely used [19,20,21,22]. Compared with conventional PCR, qPCR uses fluorescence chemical detection and computer system analysis to monitor the process of DNA amplification in real time. The fluorescent chemicals added in the amplification process can combine with specific nucleic acid sequences in order to produce brighter fluorescence or add labeled probes to specifically hybridize with DNA templates. The amplification results are collected through fluorescence chemical detection, and the output of fluorescence signal values is completed by the computer system [23].

The first commercial real-time, quantitative PCR instrument was produced by PerkinElmer Instruments Ltd. Currently, many companies are still developing real-time quantitative PCR instruments. Only hospitals, epidemic prevention stations, and scientific research institutes with sufficient funds can afford such PCR instruments [24]. Thus, PCR technology is a limited alternative for many countries. Furthermore, commercial PCR instruments on the market are bulky, expensive, and require laboratory settings, conditions which are not suitable for field operations, nor are they practical for on-site disease diagnoses in remote areas. There are numerous widely used commercial PCR instruments with excellent detection capabilities on the market. However, because of their sensitivity and signal reporting methods, real-time PCR is a technology that is ten times more expensive than standard PCR platforms [6,7,8,9]. Therefore, it is of great significance to develop a low-cost, real-time quantitative PCR instrument [25].

With these considerations in mind, we introduce a low-cost, real-time fluorescence PCR system. The overall cost of the system is about USD 760, and the system accommodates a wide variety of reaction tubes (sample holder). This system can be used to diagnose any DNA or RNA pathogen, and we provide the corresponding amplification results through the amplification of SARS-CoV-2 virus (COVID-19) as well as African swine fever virus (ASF). Simultaneously, we conducted testing trials on-site and confirmed the veracity of the results.

## 2. Materials and Methods

### 2.1. Assembly of qPCR System

#### 2.1.1. Thermocycling System

The system is composed of three subsystems: A temperature circling system, an optical detection system, and a control analysis system, as shown in Figure 1. Figure 2a depicts a comparison diagram of this system and the ABI-STEPONE system, while Figure 2b depicts a separate 45° view of this system. 

Temperature control is the most critical and essential part of the polymerase chain reaction (PCR). The function of the thermal cycling system was realized using a thermoelectric cooler (TEC; ATE1-TC-127-8ASH, Analog Technologies, Cambridge, MA, USA), several cooling fans, and a TEC-H bridge controller. The temperature control principle involves the TEC being controlled by the circuit in order to achieve the denaturation stage as well as the annealing stage. A heat-conductive metal block with a platinum thermal resistor (PT1000A, Zhengzhou, China) was placed on top of the previously mentioned TEC. For the material of the metal thermal block, we chose a low-cost aluminum alloy with high thermal conductivity. It is worth mentioning that the slot on the thermally conductive metal block is designed to be compatible with a variety of PCR reaction tubes, making the instrument more adaptable and expandable, as shown in Figure 2c.

To further reduce the total cost of the instrument, we no longer use expensive commercial thermostats and have developed a main control board that integrates PID temperature control and serial communication with the laptop, as well as controls the H-bridge module and the temperature feedback module simultaneously. The type of microprocessor we use is the ARM^®^ Cortex^®^-M3 microprocessor. For use in a cost-effective, portable PCR instrument, the microprocessor must have sufficient processing performance. An integrated development environment (IDE) or a development toolchain is also necessary for processors to be able to record from many sensors.

For the sake of facilitating the instrument’s use in distant or extreme locations, we equipped it with portable batteries as well as a reagent refrigerator. The batteries provide 24-Volt power directly to the main control board, and reagent refrigerator. Twelve-Volt power is converted to three-Volt laser power using a constant voltage module through a relay with a normally open contact. Simultaneously, when the power supply conditions permit or when battery power is insufficient, the built-in switching power supply (MEAN WELL, Guangzhou, China) can be used to provide power to the instrument. Additionally, the chosen batteries must be able to provide the energy required for at least 3 complete PCR amplification reactions. The refrigerator can maintain the reagents at 2–4 °C in order to prolong their storage life.

#### 2.1.2. Optical Feedback System

Our real-time fluorescence quantitative PCR instrument uses a laser as the fluorescence excitation source. Laser toggling is managed by the control board mentioned above. The fluorescence detectors of most commercial PCR instruments use PMT detectors (the use of real-time PCR methods in DNA sequence variation analysis), ultra-clear COMS image sensors, CCD image sensors, etc. These detectors are expensive. Researchers have employed high-definition digital cameras (Canon 7D or Canon EOS 7D, Tokyo, Japan) or used microscope cameras (E3ISPM20000KPA, Kuy Nice, Zhengzhou, China) in the past, as well as professional fluorescence cameras and cameras attached to mobile phones to gather data. In comparison to these approaches, we intended to further reduce costs while also providing sufficient sensitivity to detect fluorescence [26,27,28,29]. In order to reduce costs, we chose a USB camera equipped with a low-cost CMOS sensor. We tested five fixed-focus tiny cameras at different resolutions (1-megapixel USB camera; 2-megapixel USB camera; 5-megapixel USB camera; 8-megapixel USB camera, Shenzhen, China). Ultimately, we selected a 5-megapixel USB camera, which was sufficient for detecting fluorescent signals and cost less than USD 30.

The USB camera is packaged in a self-designed aluminum square box, with a circular narrow-band filter (520 nm) installed at the bottom of the box. This design can avoid the degradation of fluorescence detection quality caused by light leakage. In order to keep the USB camera in focus, we placed it horizontally above the heat-conductive metal, and the excitation light was tilted by 45° to illuminate the metal, as shown in Figure 2d.

#### 2.1.3. Center Controller and Software

During operation, the entire system is controlled by a laptop or a tablet. The subsystems are connected to the laptop via RS232 serial or USB ports, as shown in Figure 1. The software can run on any Windows platform. Heat-conductive metal temperature and fluorescence imaging systems can be controlled by sending commands to each subsystem from a laptop. As shown in Figure 1, a lithium battery (12 V, 12,000 mAh) was used as the power supply for the digital devices. It provides 12-Volt power directly to the PID temperature controller, bus controller, and reagent cooler. A constant voltage module and a relay with a normally open contact turns 12-Volt power into 3-Volt LED power.

The software automatically controls power supply switching. The battery is used as the power supply by default. If the integrated battery voltage feedback circuit detects insufficient battery power, it will automatically switch to an external power supply as well as provide a prompt on the software interface. A USB cable is used to link the laptop’s software to the instrument. The built-in data processing module can also store, process, and analyze the signals acquired. Troubleshooting is made easier thanks to the software’s component control and debugging features. Additionally, the reaction temperature is monitored continuously in order to minimize overheating as well as control the number of amplification cycles. There is also a temperature control function for obtaining the melting curve. The software can perform reverse transcription and pre-denaturation, as well as implement different temperature control phases.

### 2.2. Reagents

The gene segment of the SARS-CoV-2 virus was inserted into the pUC57-Kan plasmid vector (Genewiz, Suzhou, China) by recombinase and was further used as the PCR target. The reagent was composed of 4 µL Premix Ex Taq (TaKaRa Biotechnology (Dalian) Co., Ltd., Dalian, China), 1 µL EvaGreen dye (FAM channel), 1 µL forward and reverse primers, and 1 µL PCR template (Genewiz, Suzhou, China). The synthesized primer sequences for the HDLRT-qPCR system were as follows: ORF1ab-F CCC TGT GGG TTT TAC ACT TAA 8OD (forward) and ORF1ab-R ACG ATT GTG CAT CAG CTGA 8OD (reverse); N-F GG GAA CTT CTC CTG CTA GAAT 8OD (forward) and N-R CAG ACA TTT TGC TCT CA GCTG 8OD (reverse). In addition to the self-synthesized kit using the published sequence that we mentioned above, we purchased a COVID-19 rapid test kit from Daan Gene Co., Ltd. (Guangzhou, China), and used the positive control in it as well as the weak positive quality control reference material of 2019-nCoV pseudovirus RNA in oral mucous matrix (high concentration, ORF1ab: 6.8 × 10^3^ copy/mL, N: 4 × 10^3^ copy/mL) that we purchased from the China Institute of Metrology for amplification detection. This weak positive reference material simulates the matrix and virus concentration of throat swab samples in clinical tests, and can perform quality control for the entire process of throat swab sampling, from RNA extraction to nucleic acid amplification detection. The African swine fever virus fluorescent PCR nucleic acid detection kit was purchased from Beijing MingRida Technology Development Co., Ltd. (Beijing, China) The SARS-CoV-2 virus DNA quantitative diagnostic kit (Qingdao Jianma Gene Technology Co., Ltd., Qingdao, China) for on-site detection experiments consists of nucleic acid extraction reagents, nucleic acid amplification reagents, and reference standard solutions (1 × 10^3^–1 × 10^6^ copy/mL).

### 2.3. Multi-Sample Analyses

Firstly, in order to verify our system’s efficiency, we compared it to that of a quantitative real-time PCR system (STEPONE, ABI, US). We performed analysis of the gradient template amplification curve and made 10-fold serial dilutions of the pUC57-Kan plasmid vector samples as well as the African swine fever virus samples over four orders of magnitude. In order to initiate the PCR reaction, we set up thermal cycling in the software on the laptop. There were 40 cycles of 95 °C for 15 s and 60 °C for 35 s.

Secondly, real-time quantitative PCR equipment needs to have high sensitivity to ensure that no missed detections occur. In order to verify the limit of detection (LDO) of the HDLRT-qPCR system, we used the positive quality control substance in Daan Gene Co., Ltd.’s COVID-19 rapid test kit for low viral load analysis. Three concentration gradients were performed: 10^5^ copies/mL, 10^4^ copies/mL, and 10^3^ copies/mL. The amplification detection experiment was repeated three times concurrently using both the HDLRT-qPCR and ABI-STEPONE systems, and the N site (HEX channel) was identified. Thermal cycling for 2 min at 50 °C, 2 min at 95 °C, 40 cycles of 95 °C for 5 s, and 35 s at 60 °C was used. The denaturation phase as well as the annealing or extension phase were performed at 95 °C and 63 °C, respectively, requiring 40 thermal cycles of amplification procedure.

Thirdly, we extracted RNA from the weak positive quality control reference material of 2019-nCoV pseudovirus RNA in ral mucous matrix using the magnetic bead method (nucleic acid extraction kit from Shandong Brocade Diagnostic Technology Co., Ltd., Jinan, China). The viral RNA was then reverse transcribed, cDNA was synthesized, and the amplification cycle was performed, with the N and ORF1ab sites fluorescently detected using the probe method. In order to simulate detection in a clinical setting in this experiment, we resorted to the widely used COVID-19 rapid test kit from Daan Gene Co., Ltd. to detect the extracted viral RNA, thereby verifying the instrument’s clinical detection ability. The FAM channel and the HEX channel were the detection channels used.

## 3. Results and Discussion

### 3.1. Comparison of Fluorescence Detection Results between the Application of Automatic Feedback Photographing Using the HDLRT-qPCR System vs. the Commercial Quantitative Real-Time PCR System

In order to verify the gradient detection applicability of our PCR instrument, we used the developed system to initially detect synthesized DNA fragments of SARS-CoV-2. Figure 3 shows the amplification results of different concentrations of the same template in the commercial PCR instrument (Figure 3d) and in the HDLRT-qPCR system (Figure 3c).

Another set of experiments used a positive quality control substance sample from a purchased African swine fever virus (ASF) detection kit to show that other samples could be detected using the same thermal cycling system and camera, indicating the HDLRT-qPCR system’s potential for use in animal husbandry. Figure 3 shows the amplification results for different concentrations of the same template (ASF) in the commercial PCR instrument (Figure 3b) and in the HDLRT-qPCR system (Figure 3a).

The S-shaped curve reveals that the product was specific; the curve is highly recognizable, which means that this USB camera is suitable for photographing with different templates. Additionally, we compared the HDLRT-qPCR system DNA molecules with the cycle thresholds (Ct) output from the ABI-STEPONE real-time PCR system. The Ct value is the number of cycles at the point where the fluorescence threshold and the amplification curve meet.

For the SARS-CoV-2 virus (COVID-19) samples (10^6^ to 10^3^ copies/μL), the Ct values were 18.5, 21.5, 23.2, and 26.3 obtained on the HDLRT-qPCR system; meanwhile, the Ct values obtained on the ABI-STEPONE PCR instrument were 18.1, 20.2, 24.2, and 27.1. Similarly, for African swine fever virus samples (ASF), the Ct values were 23.7, 27.5, 32.3, and 36 obtained on the HDLRT-qPCR system, while the Ct values obtained on the commercial PCR instrument were 22, 26.5, 30.5, and 34.4.

By using the positive quality control material in Daan Gene Co., Ltd.’s COVID-19 rapid test kit and configuring three concentration gradients, 10^5^ copies/mL, 10^4^ copies/mL, and 10^3^ copies/mL, across three replicate tests with HDLRT-qPCR, we were able to determine the limit of detection for the HDLRT-qPCR system. As shown in Figure 4a, the system detects the HEX channel, and the results indicate that the HDLRT-qPCR system has a detection limit of 10^3^ copies/mL. Figure 4b illustrates the amplification result of the same gradient samples for the ABI-STEPONE system.

At the same time, in order to verify the detection ability of our instrument for clinical samples (RNA viruses), we performed dual-channel amplification detection on weak positive quality control reference material of 2019-nCoV pseudovirus RNA in oral mucous matrix. The amplification results of the HDLRT-qPCR system are shown in Figure 4c, while Figure 4d illustrates the amplification results of the same samples for the ABI-STEPONE system. The Ct values obtained by the HDLRT-qPCR system were 30.2, 32.8 while the Ct values obtained on the ABI-STEPONE were 29.8, 32.5.

### 3.2. Melt Curve Analysis

Currently, the majority of researchers will prioritize the fluorescent dye approach on the bases of experimental necessity and cost. The experimental design of the dye method is straightforward; only two primers are required; there is no need for probe design; the initial cost is inexpensive; and the sensitivity is high. Since fluorescent dyes bind to all double-stranded DNA, it is very important to analyze the melting curve after each reaction in order to check the specificity of the PCR assay. With the degeneration of the two-strand DNA, the fluorescent dyes would return to a free state, and this would decrease the fluorescence signal. The target melt curve should have a single peak, and the comparison of the annealing temperature of the peak position with that of the specific product can be used to determine whether the expanded product is the target product.

In the first experiment, we used EvaGreen dye (FAM channel) to detect the gene segment of the SARS-CoV-2 virus. Since the fluorescent dye method can be used to obtain the dissolution curve, we plotted melting curves by gradually increasing the temperature while monitoring the fluorescence brightness at each temperature stage. The melting temperature rises steadily from 65 °C to 95 °C, with a temperature gradient of 0.5 °C. Figure 5 shows the change in fluorescence of the product after 65 °C. Upon reaching 95 °C, the product was almost completely dissociated.

### 3.3. Field Operation

One way to scale up testing is to use a batch testing method (grouped testing or pooled testing). This process groups individual test samples together in order to speed up the whole process. This testing strategy uses batches that are between 5 and 10 samples from all around the world, only carrying out individual tests if a batch proves positive. In order to complete nucleic acid screening of large-scale populations, such detection methods are still very commonly used, and this method was used in the field detection experiments on the instruments that we developed [30].

At a conference venue, we performed an on-site nucleic acid test for the novel coronavirus. All participants in the field testing study signed a consent form after being fully informed of the purpose and methodology of the study. Oral swabs were collected at the site by professional medical staff who provided us with some of the collected samples for RNA extraction using the one-step extraction kit purchased from the previously mentioned Qingdao JianMa Gene Technology Co., Ltd. The extraction operation consisted of the following: pipette 100-microliter swab samples into a new 1.5-milliliter centrifuge tube; treat the centrifuge tube at 95 °C for 3 min; after mixing with a vortex, the solution is directly used as a template for subsequent PCR amplification. The loading sample volume is determined according to instructions from the amplification kit. We gathered nucleic acid samples from 40 volunteers prior to the meeting, used the 10-sample group testing procedure, and finished amplification testing on-site. All essential chemicals were pre-configured and stored in the instrument’s tiny refrigerator. For control testing, we added a positive control to the evaluated samples. Simultaneously, the same samples were stored, brought back to the laboratory, and the results were compared using the ABI-STEPONE instrument for comparative testing. For on-site detection experiments, the thermal cycling consisted of 55 °C for 5 min, 95 °C for 2 min, 40 cycles of 97 °C for 5 s, and 55 °C for 15 s. The test results, as shown in Figure 6, were all negative. After the meeting, all 40 participants went to the hospital to undergo nucleic acid testing for the new crown, and the hospital generated a negative nucleic acid test report. This further verifies the accuracy and reliability of the detection results produced by the equipment.

In conclusion, we have systematically verified that the constructed HDLRT-qPCR system is capable of achieving detection performance comparable to that of commercial instruments, and may be applied effectively. Some innovative and portable real-time PCR detection devices have been proposed in prior research, but they still exhibit some shortcomings. For instance, integration is relatively low, and only the detection function of a single fluorescence detection channel is specified [31]; gradient amplification analysis for multiple samples and melt curve analysis are not performed [25]. Our system’s amplification detection speed is comparable to commercial real-time PCR devices, which is one of its shortcomings. PCR devices that use heat-conductive metal, such as the system developed in this study, have slower expansion detection speeds. We use standard PCR reaction tubes rather than chip-type [31] or continuous-flow PCR devices [29,32], which require customized microtubes or chips to be set up; the reason for this is due to the fact that classic PCR reaction tubes are more prevalent and may be used more quickly.

## 4. Conclusions

This study has developed a low-cost, dual-channel, real-time fluorescent PCR instrument, with a performance that has been verified through comparison with commercial instruments; on-site detection of 2019-nCoV has also been successfully carried out. The instrument is easy to use and manufacture. The experimental results show that the instrument can meet the needs of underdeveloped and remote areas for the prevention and control of various pathogens. It does not require a laboratory environment, is convenient as well as accurate, and can complete on-site testing. Numerous websites and media outlets that cover scientific research have reported on this study. It will play an important role in the future in preventing and controlling new crown epidemics. Consequently, we believe that in the field of animal husbandry, qualification approval and application will be expedited, and that our instruments can be rapidly implemented in underdeveloped regions in order to promote economic growth of the livestock industry.

This study is not without limitations. At the moment, our system has only completed dual-channel detection; it still lacks an internal reference channel to judge the performance of the PCR reaction for SARS-CoV-2 detection. At the same time, we still need to test a significant number of positive and negative samples in a clinical setting in order to demonstrate the instrument’s sensitivity, specificity, and reproducibility. This will be a long-term stability study. In the future, we will continue to develop new multi-channel instruments to detect numerous common fluorescent signals and achieve simultaneous detection of several fluorescent sites, as well as undertake additional cost-focused research. In the future, the system will be enhanced to make it easier to operate as well as more applicable to a variety of industries.

## Figures and Tables

**Figure 1 biosensors-12-00706-f001:**
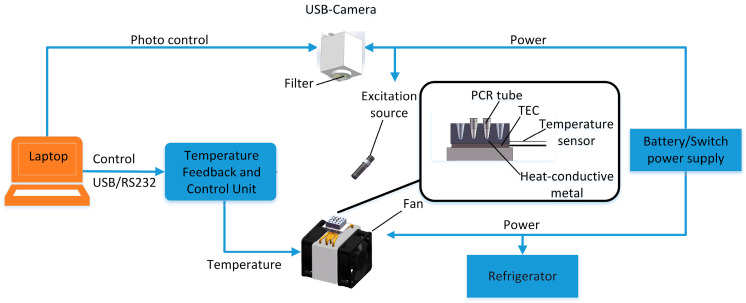
An overview of the HDLRT-qPCR system. The system can be powered by either lithium batteries or an external power supply. Micro-stepper motor-switched multiple filters in front of the USB camera are used to acquire data from multiple fluorescence channels. The temperature cycle of the thermoelectric cooler (TEC) is modulated by a control unit. The fluorescence information is collected using a USB camera.

**Figure 2 biosensors-12-00706-f002:**
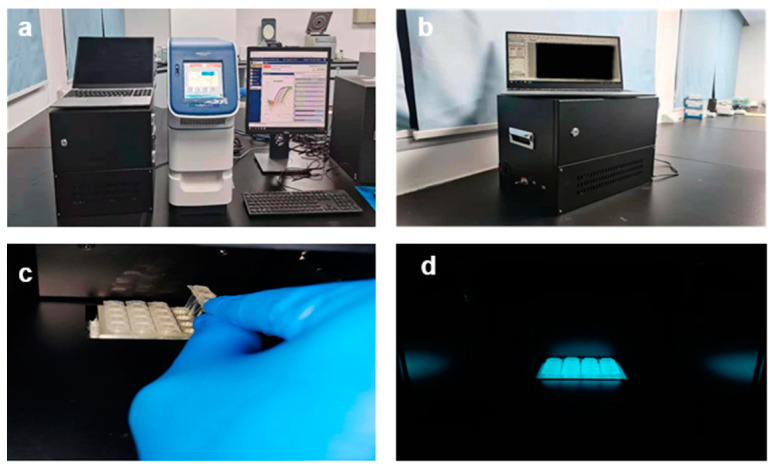
Comparison photos with (**a**) the ABI-STEPONE system and (**b**) the HDLRT-qPCR system. (**c**) The sample is placed on the heating block inside the HDLRT-qPCR system; (**d**) the light is turned on to excite fluorescence.

**Figure 3 biosensors-12-00706-f003:**
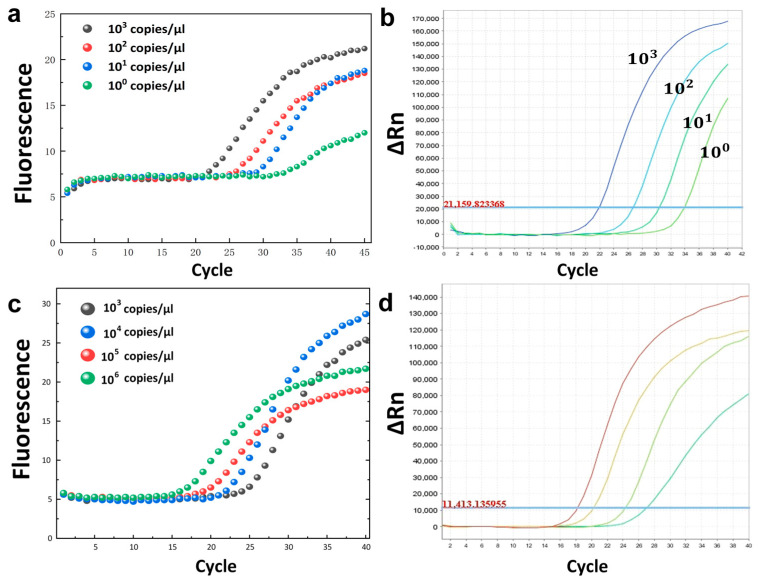
(**a**) HDLRT-qPCR gradient amplification curves for African swine fever. (**b**) African swine fever gradient amplification curves for real-time PCR amplification (ABI-STEPONE). (**c**) SARS-CoV-2 (gene target: ORF1ab) gradient amplification curves for the HDLRT-qPCR system. (**d**) ABI-STEPONE real-time PCR amplification system SARS-CoV-2 (gene target: ORF1ab) gradient amplification curves.

**Figure 4 biosensors-12-00706-f004:**
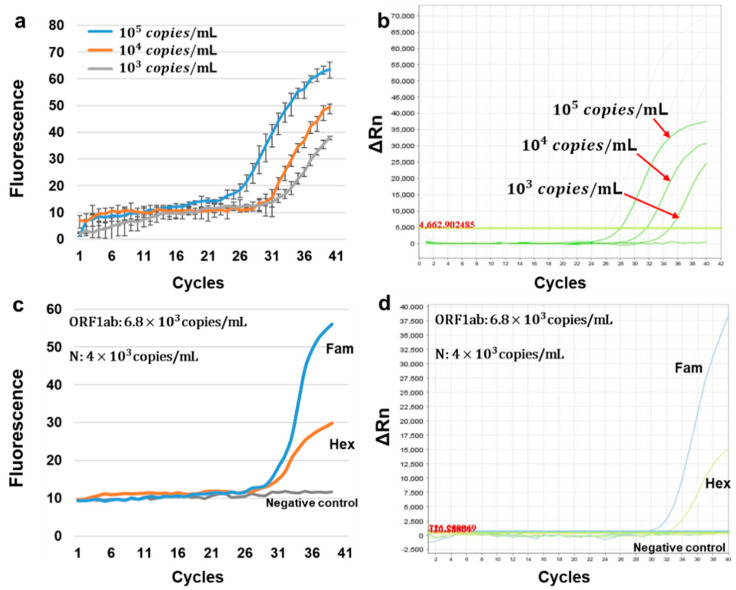
(**a**) HDLRT-qPCR gradient amplification curves for positive quality control substance from Daan Gene Co., Ltd.’s COVID-19 rapid test kit. (**b**) Positive quality control substance from Daan Gene Co., Ltd.’s COVID-19 rapid test kit amplification curves for real-time PCR amplification (ABI-STEPONE). (**c**) Weak positive quality control reference material of 2019-nCoV pseudovirus RNA in oral ucous matrix amplification curves for the HDLRT-qPCR system. (**d**) ABI-STEPONE real-time PCR amplification system weak positive quality control reference material of 2019-nCoV pseudovirus RNA in oral mucous matrix amplification curves.

**Figure 5 biosensors-12-00706-f005:**
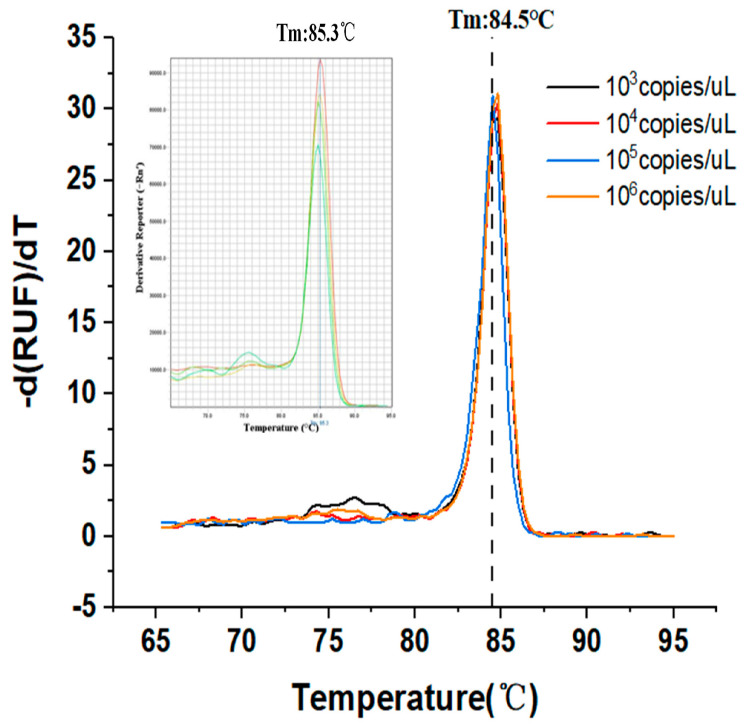
The melt curve analysis of the template with a concentration of 10^3^–10⁶ copies/µL obtained from the HDLRT-qPCR system as well as the real-time PCR amplification (ABI-STEPONE) system (shown in the upper left of the figure, the blue, green, yellow, and red lines correspond to concentrations of 10^3^–10⁶ copies/µL), showing the first derivative change in fluorescence intensity as a function of temperature. Meanwhile, only a single peak corresponding to the PCR product was observed. The amplicon is clean and specific.

**Figure 6 biosensors-12-00706-f006:**
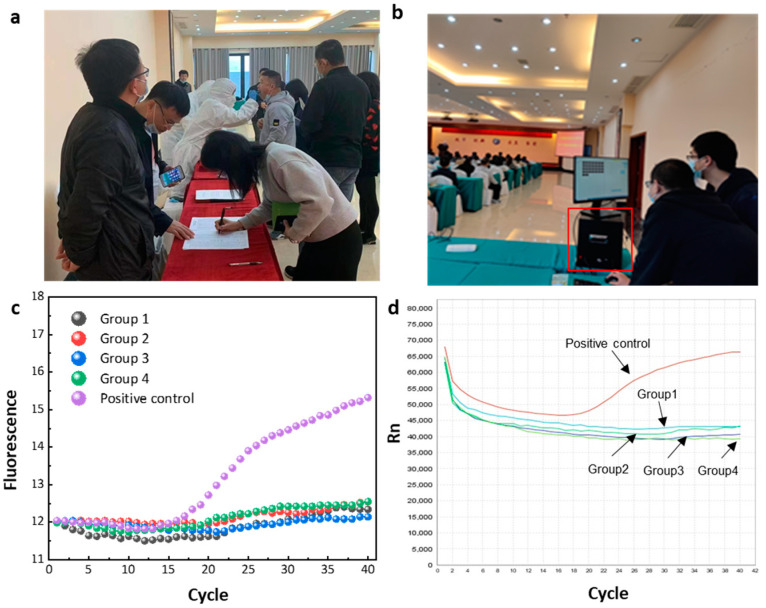
(**a**) Sampling from the conference site: upper respiratory tract nasopharyngeal swabs were used as nucleic acid detection samples. (**b**) Using the HDLRT-qPCR system (the part circled by the red frame) instrument for detection and analysis on-site. (**c**) On-site nucleic acid test results of the HDLRT-qPCR system. (**d**) Amplification curve obtained by using the ABI-STEPONE fluorescence quantitative PCR instrument for amplification detection of nucleic acid samples collected from the same batch in the laboratory. Experiments show that the amplification curve obtained by on-site sampling and analysis of the HDLRT-qPCR system is basically consistent with the curve obtained by the amplification analysis using the commercial instrument ABI-STEPONE system in the laboratory. This proves that the HDLRT-qPCR system can successfully perform field detection.

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
