# Peer review of "Hundreds-Dollar-Level Multiplex Integrated RT-qPCR Quantitative System for Field Detection"

_biosensors, 2022, doi:10.3390/bios12090706_

Round 1
Reviewer 1 Report
Review Report “Biosensors”
“Hundreds-dollar-level multiplex integrated RT-qPCR quantitative system for field detection”
Brief summary:
This manuscript focuses on a novel, cheap/low cost, fast and compact RT-qPCR instrument to apply for diagnosis in field/on site, in particular in underdeveloped regions and country. The reliability and the concordance of the method were evaluated comparing this device and tool with a commercial real-time PCR instrument, ABI-STEPONE system (Thermofisher). Amplification efficiency was tested for Sars-Cov2 AND African swine fever target genes.
Broad comments
This manuscript focuses on a novel, cheap/low cost, fast and compact RT-qPCR instrument to apply for diagnosis in field/on site, in particular in underdeveloped regions and country. The reliability and the concordance of the method were evaluated comparing this device and tool with a commercial real-time PCR instrument, ABI-STEPONE system (Thermofisher). Amplification efficiency was tested for Sars-Cov2 AND African swine fever target genes.
The originality and novelty of the manuscript is of a certain importance, because this work want to face all the problems and critical issues existing in diagnostic fields for underdeveloped country, where the resource are limited. Anyway, the manuscript has a lot of weak points and some passages in the text have to be clarified and better explained.
References in the text are not correctly reported. Follow the Biosensors guidelines, that require numbers in square brackets.
Companies of reagents and instruments have to be specified along the text, reporting also the country the first time of mention.
Please, correct some grammar and orthographical mistakes along the text. Linguistic revision is strongly recommended prior of the next resubmission.
Some specific comments:
Title and Abstract
It is not correct referring to multiplex assay, because you did not test different target genes in a same reaction or set multi-channel qPCR.
Introduction
Line 37: write in vitro in italics form.
Lines 37-ff: “it was invented by Kary Mullis in 1983….”: these sentences are unnecessary and redundant in this context.
Lines 64-66: this sentences is incomplete and not clear.
Line 85: please, update the bibliographic references. In some cases they are too long-standing.
Lines 87-89: please, reformulate this sentence because incomplete and not clear.
Lines 103-106: Why did you choose Sars Cov2 and ASF virus as pathogens to consider together? The first is well-known as still relevant worldwide issue and it is a zoonotic/pandemic agent, instead the second one interests and affects exclusively animal hosts. In particular, African Swine Fever (ASF) is a viral disease that affects pigs and wild boars. Highly contagious and often lethal to animals, it is not, however, transmissible to humans (Ministry of Health; WHO). Please better explain and contextualize.
Line 95: What do you intend with “mature commercial PCR instruments”? Please change the adjective.
Materials and method
Information about matrices, sampling and nucleic acid (DNA/RNA) extraction are completely lacking with any ethical requirements. They are of fundamental importance, so please add a paragraph describing in detail this experimental section.
2.1 Assemly of qPCR system
Please better specify and highlight that this system was realized and performed by your research group: it is the real strength point of this work. Anyway I suppose that a qPCR instrument even more small, laptop and “easy to use” directly in field represents the ambitious goal, possibly combined to an extraction system.
2.1.2 Optical feedback system
Figure 2: please add the letters a-b-c-d near to the figures/images.
Figure 3: this figure is not informative and it is superfluous. Please, remove it.
2.2 Reagents
Please, better specify along the text the gene targets for Sars-Cov2 and for ASF virus. Were the primer ORF1ab designed by you or taken from the literature?
2.3 PCR amplification
Please change the title of this paragraph.
Line 202: the concordance of the two PCR instruments, as well as applicability of the method, has to be verified.
Line 203: How many replicates of dilution have you prepared?
You reported that 10-fold serial dilutions were performed, but actually only five dilutions were reported also in “Results section”. You referred to four dilutions point (106 cg/µl up to 103 cg/µl). Furthermore, the last dilution is too high in terms of genomic copies for microliter, in order to define the sensibility of a qPCR method. If it is possible, I would suggest setting up more dilutions points in at least 3 replicates and verifying the LOD (limit of detection) and the cut-off of the systems, always comparing the two tools. It is fundamental that this device and assay reveal also traces and low viral load, whether it is Sars-Cov2 or ASF virus. It would be desirable also evaluate and test specificity (inclusivity and exclusivity panel) and robustness of the method.
Lines 204-206: information about thermal conditions are redundant and repeated twice.
Results and discussion
This section, in particular discussion section, has to be implemented with updated and recent references in pertinent field, if possibile.
Line 215 and along the text: please, write “Fig” in full length.
Lines 218-221: please, try to explain and better clarify this concept
Lines 221-229: this paragraph has to be moved into Introduction, adding some information about Sars-Cov2 (COVID-19) and African swine fever virus.
Lines 225-226: please clarify this statement. The temperature should be uniform in each well of PCR thermo-block.
Figure 4: please add the letters a-b-c-d near to the figures/images and better specify the gene targets that are not inferred from the figures. Anyway, ∆Rn value is lower for HDLRT-PCR compared to ABI-STEPONE, so the amplification efficiency is minor.
3.2 Melt curve analysis.
Please, try to explain better this paragraph. In my opinion, the concept reported in lines 258-263 is too detailed and misleading.
3.3 Field operation
Lines 270-276: please, try to explain better this paragraph because not clear and misleading.
As mentioned before, please report RNA extraction method from swabs samples.
References
Better check the reported references also in accordance with the format required by “Biosensors-MDPI”.
Author Response
Dear Editor and Reviewers:
Thank you for taking out of your busy schedule to review the manuscript. We are very grateful to Reviewer for reviewing the paper so carefully. We have carefully considered the suggestion of Reviewer and make some changes.
Please do not hesitate to contact us if there are any question. Thanks again to the reviewers and enditors for your hard work! Best wishes for you!

Reviewer 2 Report
The authors in this article describe the development of a low-cost single channel real time instrument that can potentially be used in field detection of infectious agents such as SARS-CoV-2 and African swine fever virus.
As the idea is very useful and helpful in the field of detection of infectious diseases and on site point of care testing three major concerns are needed to be addressed:
1. As the instrument is a homemade instrument, the authors need to provide evidence that the long time use of the instrument will result in reliable and concordant results.
2. As the instrument is a single channel detection system, this makes unsuitable for the detection of SARS=COV-2 since the recommendation of the WHO require the detection of two gene targets. So, the use of this instrument in detecting COVID cases will result in prolonged time from sampling to result, besides it will not allow for the use of internal control to judge the performance of the PCR reaction.
3. The authors also used a DNA template for the detection of SARS-CoV-2 which is not the normal template in clinical samples as the virus is an RNA virus, so the validation of the assay is not applicable. They also need to test a large number of positive and negative samples to show the sensitivity, specificity and reproducibility of the instrument.
Minor comments:
1. Figure 2 is better be a diagram showing the concept of the instrument than showing photos of the machines.
2. Page 5 line 177: the authors state that “there is a button that allows the user to choose between battery power and external power”, technically this needs to be automatic switch to avoid human error when running the instrument and forgetting to switch to external power.
3. Page 5 lines 191-192: the authors used a gene segment DNA of SARS-CoV-2 “The gene segment of the SARS-CoV-2 virus was inserted into the pUC57-Kan plasmid vector (Genewiz, Suzhou, China) by recombinase and further used as the PCR target” as the SARS-CoV-2 virus is an RNA, this template DNA is not the proper template for evaluating the performance of the assay.
4. Page 6 lines 205-206: the authors describe the PCR amplification using a SARAS-CoV-2 sample while the PCR thermal protocol does not include an RT step. If the SARS sample is DNA then it is not the appropriate sample type. There needs to be more elaboration on the chemistry here not just a general description of the cycling steps which refers to a PCR reaction not RT-PCR reaction .
5. Page 8 Lines 279-283: procedure for sample collection extraction and amplification is needed. While this experiment results shows the performance of the instrument in negative samples, testing of positive samples are needed to know the performance of the instrument on real clinical samples.
6. In the conclusion section, the authors state that the instrument is a single channel PCR which makes its use in detecting COVID cases needs two runs to amplify two SARS-CoV-2 targets as recommended by the WHO. This will increase the time needed from sampling to result as it will not be able to perform multiplex PCR to detect the virus in one run and will not be able to add an internal control to control for the PCR efficiency and potential inhibition.
7. Page 2 line 59: the authors stated “skipping the link between sample circulation and laboratory testing” which is not related to the context.
8. The authors stated in several occasions the novel coronavirus, this needs to be changed to SARS-CoV-2.
9. I am not sure about the policy of the journal concerning ethical approval, but this study needs to be covered by an ethical approval from an IRB. The consent of the participants is not enough without IRB.
Author Response

(The authors gave the same response as above.)

Round 2
Reviewer 1 Report
Dear Authors, you have improved the quality of the manuscript implementing also the experimental design.
Anyway I have still some doubts.
Firstly, English revision is required .
Furthermore you stated and supported that HEX channel for N gene was added, but probe sequence, data and amplification plot are not present in the main text.
What kind of matrix is the Weak Positive Quality Control Reference Material of 2019-nCoV Pseudovirus RNA 287 in Oral Mucous Matrix from which you have extracted the Sars-Cov2 RNA? It is not clear. Is it an oral, rhinal-pharyngeal swab or what else?
Lines 213-217: please, reformulated this misleading passagew
Lines 250-255: the minor Cq should correspond to the higher copy number and not vice versa.
Results and discussion section have to be implemented and discussed with references.
Please, pay attention to acronimous, for example for Real-time PCR. You have to report them in entire form the first time, explaining the acronimous used the following times.
Thanks for your consideration.
Author Response
1.We have revised the language section.
2.We have supplemented the probe sequences for the N gene, plese refere line 198-196 The data and amplification plot please refere Figure 4(c).
3.This weak positive reference material simulates the matrix and virus concentration of throat swab samples in clinical tests, and can perform quality control on the whole process of throat swab samples from RNA extraction to nucleic acid amplification detection. And the standard material is the NIM-RM5203 novel coronavirus (2019-nCoV) pseudovirus RNA standard material developed by the Chinese Academy of Metrology, added to simulated oral mucus containing alpha-amylase, lysozyme, mucin, and artificial saliva. In the matrix, a high concentration of weakly positive novel coronavirus pseudovirus oral mucus matrix solution was prepared, covering the novel coronavirus nucleocapsid protein N gene (full length) and the open reading frame 1ab (ORF1ab) gene fragment (the genome coordinates are: 13201-15600 and 18500-19000, GenBank No.NC_045512).
4. We are very sorry for our negligence in the explanation. We have rewritten lines 213-217.Thanks for your correction.We have corrected the writing order.
5. We supplement the proposed devices with some references in this section, and compare and analyze the advantages and disadvantages of the devices. Line 358-368.
6. We have made corrections in the manuscript.